# Exploring spatial variations and the individual and contextual factors of uptake of measles-containing second dose vaccine among children aged 24 to 35 months in Ethiopia

**Achamyeleh Birhanu Teshale** [1] *, **Tsegaw Amare** [2]

**1** Department of Epidemiology and Biostatistics, Institute of Public Health, College of Medicine and Health Sciences, University of Gondar, Gondar, Ethiopia, **2** Department of Health Systems and Policy, Institute of Public Health, College of Medicine and Health Sciences, University of Gondar, Gondar, Ethiopia

* achambir08@gmail.com

## Abstract

### Background

To eliminate measles, which is a devastating contagious disease, Ethiopia introduced the measles-containing second dose vaccine (MCV2) that will be given in the second year of life. Despite its paramount benefit, the coverage was low worldwide and, in Ethiopia, there is scarce evidence on the geographic variations and factors associated with uptake of MCV2.

### Objective

This study aimed to explore the spatial variations and the individual and contextual factors of uptake of measles-containing second dose vaccine among children aged 24 to 35 months in Ethiopia.

### Methods

We used the 2019 Ethiopian Mini Demographic and Health Survey data. A total weighted sample of 800 children aged 24 to 35 months was used. Multilevel analysis was employed and an adjusted odds ratio (AOR) with a 95% confidence interval (CI) was reported. Factors with a p-value<0.05 in the multivariable analysis were declared to be significant predictors of MCV2 uptake. To explore the spatial variations of MCV2 uptake, we have conducted spatial analysis using both Arc GIS version 10.7 and SaTScan version 9.6 software.

### Results

The proportion of MCV2 uptake was 9.84% (95% CI: 7.96%, 12.11%). Children whose mothers were aged 20–34 years (AOR = 0.19; 95%CI: 0.05, 0.69) and 35–49 years (AOR = 0.21; 95%CI: 0.04, 0.90), being the 4th-5th child (AOR = 4.02; 95%CI: 1.45, 11.14) and 6th and above child (AOR = 4.12; 95%CI: 1.42, 13.05) and children who did not receive full

**Data Availability Statement:** All relevant data are within the manuscript.

**Funding:** The author(s) received no specific funding for this work.

**Competing interests:** The authors have declared that no competing interests exist.

**Abbreviations:** ANC, Antenatal Care Visit; AOR, Adjusted Odds Ratio; CI, Confidence Interval; CSA, Central Statistical Agency; EAs, Enumeration Areas; EMDHS, Ethiopia Mini Demographic and Health Survey; ICC, Intra-class correlation coefficient; LLR, Log-Likelihood Ratio; MCV2, Second dose of Measles Containing Vaccine; MOR, Median odds ratio; PCV, Proportional Change in Variance; PNC, Postnatal Care; RR, Relative Risk; SNNPR, Southern Nation Nationalities and Peoples Region; WHO, World Health Organization.

childhood vaccinations (AOR = 0.44; 95%CI: 0.25, 0.77) were significantly associated with MCV2 uptake. Besides, MCV2 uptake was clustered in Ethiopia (Global Moran's I = 0.074, p-value <0.01). The primary cluster spatial window was detected in the Benishangul-Gumuz region with LLR = 10.05 and p = 0.011.

## Conclusion

The uptake of MCV2 in Ethiopia was low. Maternal age, birth order, and uptake of the other basic vaccines were associated with MCV2 uptake. Besides, MCV2 uptake was clustered in Ethiopia and the primary cluster spatial window was located in the Benishangul-Gumuz region. Therefore, special concern should be given to regions with lower MCV2 uptake such as the Benishangul-Gumuz region. Besides, it is better to give attention to basic vaccination programs.

## Background

One of the major causes of death among under-five children is measles, which is a very contagious respiratory disease caused by the measles virus. It spreads through respiratory droplets when an infected person coughs or sneezes [1, 2]. Globally, according to estimates from the World Health Organization (WHO) and the United States Centers for Diseases Control and Prevention, more than 140,000 people died from measles in 2018 [3].

In Ethiopia, the incidence of measles is high (greater than 50 cases per 1,000,000 population per year) [4] despite the childhood immunization coverage is improved through the combined effect of reaching every district approach, health extension program, and implementation of enhanced routine immunization activities [3–6]. Measles outbreaks are occurring continuously in most parts of the country as well [5].

The Global Measles and Rubella Strategic Plan 2012–2020 was developed by WHO or United Nations Children's Fund and other partners [7]. The main goal of this strategic plan was to give each child measles-containing second dose vaccine (MCV2) [8]. To eliminate this devastating contagious disease, Ethiopia also introduced the MCV2 on February 11, 2019 [4]. The uptake of the MCV2 vaccine helps to prevent measles outbreaks, attain the measles elimination goals, and resolve primary vaccination failure or boost the antibody titers after the immunity wanes. Besides, adding MCV2 to the national immunization schedule can be helpful to provide additional health services [4].

In Ethiopia, the national targets for measles accelerated control by 2012 (less than five cases per 1,000,000 population per year), and its elimination by 2020 (less than one case per 100,000 population per year) was not achieved [4]. Different studies revealed different factors for the uptake of MCV2 such as maternal education, age of the mother, wealth status, region, recommended antenatal care visit (ANC), place of delivery, postnatal care (PNC), distance from the health facility, and residence [9–14].

Despite the paramount benefit of MCV2, there is scarce evidence regarding the geographic variations and the factors associated with the uptake of MCV2. Therefore, this study aimed to explore the spatial variations and the individual and contextual factors of uptake of measles-containing second dose vaccine among children aged 24 to 35 months in Ethiopia.

## Methods

### Data source

The 2019 Ethiopian Mini Demographic and Health Survey (EMDHS), the second EMDHS, that was implemented from March 21 to June 28, 2019, was our data source. It was conducted by the Ethiopian Public Health Institute in collaboration with the Central Statistical Agency (CSA) and the Federal Ministry of Health. The survey is intended to generate data for assessing the progress of maternal and child health.

### Study population and sampling procedure

A complete list of the 149,093 census enumeration areas (EAs) created for the 2019 Ethiopian Population and Housing Census that was conducted by the CSA was used as a sampling frame. The survey used a two-stage stratified sampling. At stage one, 93 urban EAs and 212 rural EAs (total EAs = 305) were chosen with a probability proportional to EA size. Then, a household listing operation was carried out in all the selected EAs from January to April 2019 and the resulting lists served as a sampling frame for the selection of households. Then, in the second stage, a fixed number of 30 households per cluster were selected from the newly created household lists.

Among the different datasets in the survey, we used the Kid's data set (KR data). Our study population was children aged from 24 to 35 months. If there is more than one child in the age range, we took the last child since information such as ANC and other maternal characteristics are for the last child. Finally, for this study, a total weighted sample of 800 (unweighted = 809) children aged 24 to 35 months was used. Further information regarding the survey was found elsewhere [15, 16].

### Study variables

**Outcome variable.**   MCV2 uptake was the outcome variable, a binary outcome variable recoded as "1" if the child received the vaccine and "0" if the child did not receive the vaccine. Written vaccination records such as vaccination cards and mothers' verbal reports were used to identify whether the child receives the vaccine.

**Independent variables.**   Incorporates both individual and community-level variables. The individual-level variables were the age of the mother (recoded as 15–19, 20–34, and 34–49 years), educational status of the mother (categorized as no formal education, primary education, and secondary and above), marital status (currently married or unmarried), religion (categorized as Orthodox Christian, Protestant, Muslim, and others), wealth index (with four categories; poorest, poorer, middle, richer, and richest), sex of household head (male or female), birth order (recoded as only one, 2–3, 4–5, and 6 and above), possession of radio (yes or no), possession of television (yes or no), recommended ANC visit (yes or no), place of delivery (home or health facility), sex of the child (male or female), taking full immunization for other childhood vaccinations (yes or no), and child PNC checked within 2 months (yes or no). While place of residence (urban or rural) and region (large central, small peripheral, or metropolitan) were the community-level variables.

### Operational definitions

**Region.**   Ethiopia has nine geographical regions and two administrative cities (a total of 11 regions), for this study, we have categorized these regions into three since there were small samples per some of the regions; the large central regions (Tigray, Amhara, Oromia and Southern Nation Nationalities and Peoples Region (SNNPR)), the small peripheral regions

(Afar, Somalia, Gambela, and Benishangul-Gumuz region), and the metropolitans (Dire Dawa, Addis Ababa, and Harari). This categorization is in line with different studies conducted elsewhere [17, 18].

**Full immunization for basic vaccines.** We considered full vaccination if the child received one dose of BCG, three doses of pentavalent, three doses of PCV, three doses of OPV, and two doses of Rotavirus. Then the immunization status was recoded as "yes" if the child received all vaccines and "no" if at least one vaccine or dose was missed. Taking the measles first dose vaccine at nine months is excluded deliberately because to take the MCV2, it is a must to take the first dose.

**Recommended ANC visit.** If a woman had at least four ANC visits during her pregnancy for the child incorporated in this study.

## Data management and statistical analysis

The data were managed (extracted, recoded, and cleaned) using Stata version 16. Description of study participants and the proportion of uptake of MCV2 with its 95% confidence interval (CI) were reported using text and tables.

**Multilevel analysis of the uptake of MCV2.** *Fixed effects analysis (measures of association).* We have employed a multilevel multivariable logistic regression analysis since the survey had hierarchical nature (children were nested within EAs). Four models were fitted initially; a model with no explanatory factors (Null model), a model with only individual-level factors (Model I), a model with community-level factors (Model II), and a model fitted with both individual and community-level factors (Model III). For selecting the best fit model, we have used deviance. Then, an adjusted odds ratio (AOR) with a 95% confidence interval (CI) was reported for all models. Finally, variables with a P-value <0.05 in the best-fitted model were declared to be significant predictors for MCV2 uptake.

*Measures of variation (random-effects).* we have employed intra-class correlation coefficient (ICC), median odds ratio (MOR), and proportional change in variance (PCV) to assess variations in MCV2 uptake between or across clusters [19–21].

**Spatial analysis of the uptake of MCV2.** For the spatial analysis (spatial autocorrelation, interpolation, hot spot analysis, and SaTScan analysis), Arc GIS version 10.7 and Kuldorff's SaTScan version 9.6 software were used. To ascertain whether the spatial distribution of uptake of MCV2 was clustered, dispersed, or random across the study area (Ethiopia), the global spatial autocorrelation using the Global Moran's I statistic was employed [22]. Spatial interpolation was employed to predict the proportion of uptake of MCV2 on the un-sampled areas based on the sampled measurements [23]. Besides, using Getis-Ord Gi* statistics, hot spot regions (regions with lower rates of MCV2 uptake) and cold spot regions (regions with higher rates of MCV2 uptake) were identified [22, 24]. Furthermore, to detect significant primary and secondary clusters, the Bernoulli-based spatial scan statistical analysis was employed [25]. While conducting the analysis, our cases were children with non-uptake of MCV2 and our controls were children with the uptake of MCV2. The coordinate files (latitude and longitude) were also used since they were necessary during the analysis. To identify both small and large clusters and missed clusters containing more than the maximum limit, the maximum spatial cluster size of less than fifty percent of the population as the upper limit was used. The log-likelihood ratio (LLR) test with its p-value was reported. The likelihood function is maximized across all window locations and sizes, with the most likely cluster being the one with the highest likelihood. This is the cluster with the least likelihood of occurring by chance. The p-value is determined by comparing the rank of the maximum likelihood from the real data set to the maximum likelihood from the random data sets using Monte Carlo hypothesis testing [16].

Besides, the relative risk (RR) of uptake of MCV2 in a specific spatial window was calculated. The spatial window with the highest LLR test was considered the most likely cluster (primary cluster), a window with the lowest uptake of MCV2.

### Ethical consideration

This study was conducted under the Declaration of Helsinki and since we were using publicly accessible data, ethical approval was not needed. However, by registering or online requesting, we accessed the data set from the DHS website (https://dhsprogram.com) and the dataset had no personal identifiers.

## Results

### Characteristics of study participants

Nearly three fourth of the study participants were in the age group 20–34 years with a mean age of 28 (SD±6.41) years. The majority (56.01%) of the respondents had no formal education and 94.20% of the participants were married during the survey. The majority of the respondents were from male-headed households (88.86%) and from households that did not possess radio (73.20%) and television (86.43%). Regarding recommended ANC visit, place of delivery, and baby's postnatal checkup within 2 months, 59.84%, 52.21%, and 83.50% of the study participants had no recommended ANC visit, gave birth at home, and reported that the baby had checked within 2 months of delivery respectively. Only 44.86% of children had complete immunization for basic vaccines. The majority of the participants were rural dwellers (73.89%) and were from large central regions (88.06%) (Table 1).

### The proportion of uptake of MVC2 in Ethiopia

In this study, based on maternal reports and using the vaccination card, the proportion of uptake of MCV2 was 9.84% (95%CI: 7.96%, 12.11%) (Fig 1).

### Factors associated with uptake of MCV2 in Ethiopia

**Fixed effect analysis.**   In the multilevel multivariable analysis, maternal age, birth order, and full immunization for the basic childhood vaccinations were significantly associated with MCV2 uptake. The odds of uptake of MCV2 were 81% (AOR = 0.19; 95%CI: 0.05, 0.69) and 79% (AOR = 0.21; 95%CI: 0.04, 0.90) lower among mothers in the age group 20–34 and 35–49 years as compared with young age mothers (aged 15–19 years) respectively. Being the 4th-5th and 6th and above child had 4.02 (AOR = 4.02; 95%CI: 1.45, 11.14) and 4.12 (AOR = 4.12; 95%CI: 1.42, 13.05) times higher odds of MCV2 uptake as compared to being the first child. Children who did not receive basic childhood vaccinations were 56% (AOR = 0.44; 95%CI: 0.25, 0.77) less likely to receive MCV2 as compared to their counterparts (Table 2).

**Random effect analysis.**   The ICC in the null model revealed that about 10.8% of the variability in the uptake of MCV2 was attributed due to the difference between clusters. Besides, the MOR in the null model revealed that if we randomly select a child in two different clusters, the child from a cluster with higher uptake of MCV2 had 1.82 times higher odds of receiving MCV2 as compared with a child from a cluster with lower uptake. Besides, in model III the PCV was highest (88.2%), which revealed that model III best explains the variability of uptake of MCV2. Furthermore, model III was selected as the best-fitted model since it had the lowest deviance (Table 3). Therefore, all the interpretations, in this study,

**Table 1. Characteristics of study participants.**

| Variables | Frequency (N = 800) | Percentage |
|---|---|---|
| Maternal age | | |
| 15–19 | 27 | 3.38 |
| 20–34 | 590 | 73.79 |
| 35–49 | 183 | 22.83 |
| Highest educational level | | |
| No education | 448 | 56.01 |
| Primary | 256 | 31.95 |
| Secondary & above | 96 | 12.04 |
| Religion | | |
| Orthodox | 269 | 33.66 |
| Protestant | 252 | 31.52 |
| Muslim | 269 | 33.62 |
| Other* | 10 | 1.21 |
| Marital status | | |
| Married | 753 | 94.20 |
| Unmarried | 47 | 5.80 |
| Wealth | | |
| Poorest | 169 | 21.19 |
| poorer | 177 | 22.10 |
| middle | 170 | 21.30 |
| richer | 128 | 16.01 |
| richest | 156 | 19.40 |
| Sex of Household head | | |
| Male | 711 | 88.86 |
| Female | 89 | 11.14 |
| Household has television | | |
| No | 691 | 86.43 |
| Yes | 109 | 13.57 |
| Household has radio | | |
| No | 585 | 73.20 |
| Yes | 215 | 26.80 |
| Birth order | | |
| 1 | 156 | 19.45 |
| 2–3 | 225 | 28.09 |
| 4–5 | 195 | 24.39 |
| 6&above | 224 | 28.07 |
| ANC visit | | |
| No | 479 | 59.84 |
| Yes | 321 | 40.16 |
| Place of delivery | | |
| Home | 418 | 52.21 |
| Health Facility | 382 | 47.79 |
| Baby postnatal check within 2 months | | |
| No | 668 | 83.50 |
| Yes | 132 | 16.50 |
| Sex of child | | |
| Male | 424 | 53.04 |

(*Continued*)

**Table 1.** (Continued)

| Variables | Frequency (N = 800) | Percentage |
|---|---|---|
| Female | 376 | 46.96 |
| Full immunization for other vaccines | | |
| Yes | 359 | 44.86 |
| No | 441 | 55.14 |
| Place of residence | | |
| Urban | 209 | 26.11 |
| Rural | 591 | 73.89 |
| Region | | |
| Large central | 704 | 88.06 |
| Small peripheral | 62 | 7.72 |
| Metropolitan | 34 | 4.23 |

Note: * = traditional, catholic, and other religions

were based on model III, which incorporates both individual and community level factors simultaneously.

## Spatial analysis of MCV2 uptake

**Spatial autocorrelation.** The global spatial autocorrelation result revealed that MCV2 uptake was clustered in Ethiopia (Global Moran's I = 0.074, p-value <0.01) (Fig 2).

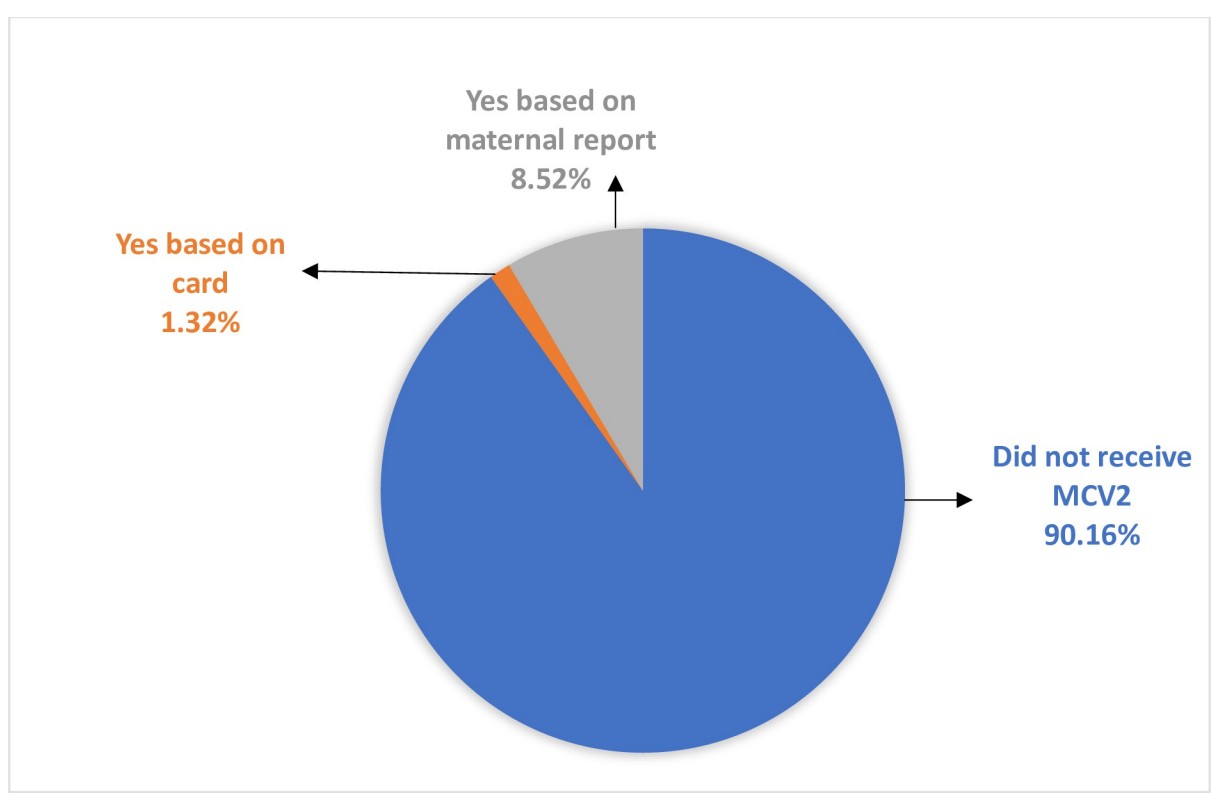

**Fig 1. Proportion of the uptake of measles-containing second dose vaccine among children aged 24 to 35 months in Ethiopia.**

**Table 2. Multilevel analysis for assessing factors associated with uptake of MCV2 in Ethiopia.**

| Variables | Null model | Model 1 AOR (95%CI) | Model 2 AOR (95%CI) | Model 3 AOR (95%CI) |
|---|---|---|---|---|
| Maternal age | | | | |
| 15–19 | | 1.00 | | 1.00 |
| 20–34 | | 0.19 (0.05, 0.73) | | 0.19 (0.05, 0.69) * |
| 35–49 | | 0.21 (0.04, 0.94) | | 0.21 (0.04, 0.90) * |
| Highest educational level | | | | |
| No education | | 1.00 | | 1.00 |
| Primary | | 0.72 (0.37, 1.42) | | 0.74 (0.38, 1.44) |
| Secondary & above | | 1.97 (0.80, 4.84) | | 2.12 (0.86, 5.19) |
| Marital status | | | | |
| Married | | 1.00 | | 1.00 |
| Unmarried | | 1.13 (0.38, 3.33) | | 1.14 (0.39,3.32) |
| Wealth | | | | |
| Poorest | | 1.00 | | 1.00 |
| Poorer | | 1.88 (0.85, 4.18) | | 1.71 (0.77, 3.81) |
| Middle | | 2.10 (0.88, 5.01) | | 1.85 (0.77, 4.47) |
| Richer | | 1.91 (0.72, 5.05) | | 1.76 (0.65, 4.72) |
| Richest | | 0.88 (0.21, 3.79) | | 0.82 (0.16, 4.08) |
| Religion | | | | |
| Orthodox Christian | | 1.00 | | 1.00 |
| Protestant | | 0.92 (0.45, 1.89) | | 0.96 (0.47,1.95) |
| Muslim | | 0.76 (0.40, 1.45) | | 0.88 (0.45, 1.75) |
| Other | | 2.21 (0.37, 13.30) | | 2.29 (0.40, 13.12) |
| Sex of household head | | | | |
| Male | | 1.00 | | 1.00 |
| Female | | 1.40 (0.68, 2.88) | | 1.54 (0.74, 3.20) |
| Household has television | | | | |
| No | | 1.00 | | 1.00 |
| Yes | | 1.36 (0.33, 5.52) | | 1.39 (0.34, 5.73) |
| Household has radio | | | | |
| No | | 1.00 | | 1.00 |
| Yes | | 0.57 (0.28, 1.14) | | 0.57 (0.28, 1.13) |
| Place of delivery | | | | |
| Home | | 1,00 | | 1.00 |
| Health facility | | 1.23 (0.66, 2.28) | | 1.24 (0.67, 2.30) |
| Birth order | | | | |
| 1 | | 1.00 | | 1.00 |
| 2–3 | | 2.16 (0.83, 5.64) | | 2.24 (0.86, 5.84) |
| 4–5 | | 3.90 (1.41, 10.83) | | 4.02 (1.45, 11.14) * |
| 6 and above | | 4.22 (1.38, 12.83) | | 4.12 (1.42, 13.05) * |
| Sex of the child | | | | |
| Male | | 1.00 | | 1.00 |
| Female | | 1.20 (0.71, 2.03) | | 1.20 (0.71, 2.02) |
| Baby postnatal check within 2 months | | | | |
| No | | 1.00 | | 1.00 |
| Yes | | 1.30 (0.68, 2.47) | | 1.26 (0.67, 2.39) |
| Full immunization for other vaccines | | | | |
| Yes | | 1.00 | | 1.00 |

(*Continued*)

**Table 2.** (Continued)

| Variables | Null model | Model 1 AOR (95%CI) | Model 2 AOR (95%CI) | Model 3 AOR (95%CI) |
|---|---|---|---|---|
| No | | 0.44 (0.25, 0.77) | | 0.44 (0.25, 0.77) ** |
| Place of residence | | | | |
| Urban | | | 1.00 | 1.00 |
| Rural | | | 1.01 (0.51, 2.03) | 1.02 (0.39, 2.66) |
| Region | | | | |
| Large central | | | 1.00 | 1.00 |
| Small peripheral | | | 0.58 (0.32, 1.04) | 0.64 (0.33, 1.26) |
| Metropolitan | | | 0.62 (0.28, 1.35) | 0.80 (0.32, 2.02) |

Note: ** = p<0.01 and

* = p<0.05

**Spatial interpolation.** The kriging interpolation result revealed that Somalia, Benishangul, western Oromia, and central and southeastern Afar regions had a higher predicted proportion of non-uptake of MCV2. However, Gambela, SNNPR, the border between Amhara and Tigray, the border between Afar and Oromia, and northeastern Oromia had a lower predicted proportion of non-uptake of MCV2 (Fig 3).

**Hot spot and cold spot analysis (Getis-Ord Gi\*).** As shown in blue-coloured cluster points, Gambela, SNNPR, and the Northern end of the Amhara region were the cold spot areas (areas with higher rates of MCV2 uptake). While Benishangul-Gumuz was the hotspot region (a region with lower rates of MV2 uptake) (Fig 4).

**SaTScan analysis.** The SaTScan analysis detected a total of 27 significant clusters and all of them were primary clusters. The most likely SaTScan cluster (primary cluster) was detected in the Benishangul-Gumuz region with LLR = 10.05 and p = 0.011, centered at 5.856584 N, 43.726016 E with 402.89 km radius, and a RR of 1.11. This revealed that children within the spatial window had 1.11 times higher odds of non-uptake of MCV2 as compared to those children outside the spatial window (Fig 5).

## Discussion

In this study, we assessed the spatial variation and factors associated with the newly introduced MCV2 uptake among children aged 24 to 35 months in Ethiopia. The study showed that 9.84% (95% CI: 7.96%, 12.11%) of children took MCV2. It is far from the plan to reach 80% of MCV2 coverage by the end of 2025 and 68.1% of global MCV2 vaccination coverage [26]. The finding is also lower than the coverage in Guinea Bissau [27], Lebanon, and China where 93%, 60.9%, and 93.9% of children have been vaccinated for MCV2. The discrepancy in the MCV2

**Table 3. Variability (random effect analysis) of uptake of MCV2 in Ethiopia.**

| Parameter | Null model | Model 1 | Model 2 | Model 3 |
|---|---|---|---|---|
| Variance (SE) | 0.397 (0.473) | 0.167 (0.474) | 0.302 (0.456) | 0.047 (0.473) |
| ICC | 10.8% | 4.8% | 8.4% | 1.4% |
| PCV | Reference | 57.9% | 23.9% | 88.2% |
| MOR | 1.82 | 1.47 | 1.69 | 1.23 |
| LL | -247.07 | -226.67 | -245.03 | -223.83 |
| Deviance | 494.14 | 453.34 | 490.06 | 447.66 |

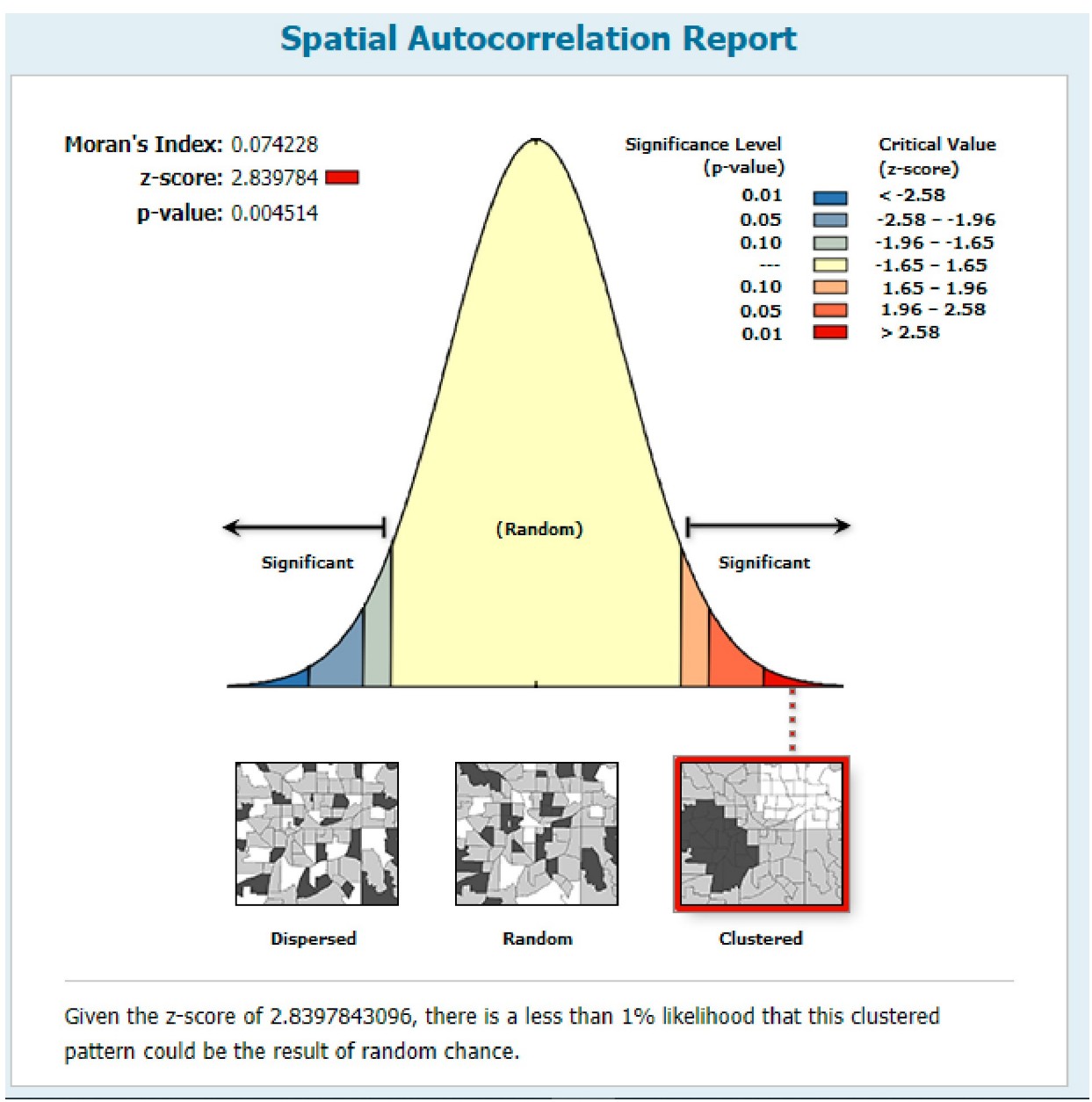

**Fig 2. Spatial autocorrelation showing the clustering of MCV2 uptake, Map produced using Arc GIS version 10.7.**

uptake might be due to the difference in the period of introduction of the vaccine and sociocultural characteristics of the respondents.

In the multivariable multilevel analysis maternal age, birth order, and full immunization for basic vaccines were significantly associated with the uptake of MCV2. Consistent with the studies conducted on the uptake of the first dose of the measles vaccine and other vaccines [28, 29], children from older mothers were less likely to get vaccinated as compared with children from younger mothers. This might be because older women are less likely than younger women to use different sources of information such as social media and learn about a newly launched vaccine.

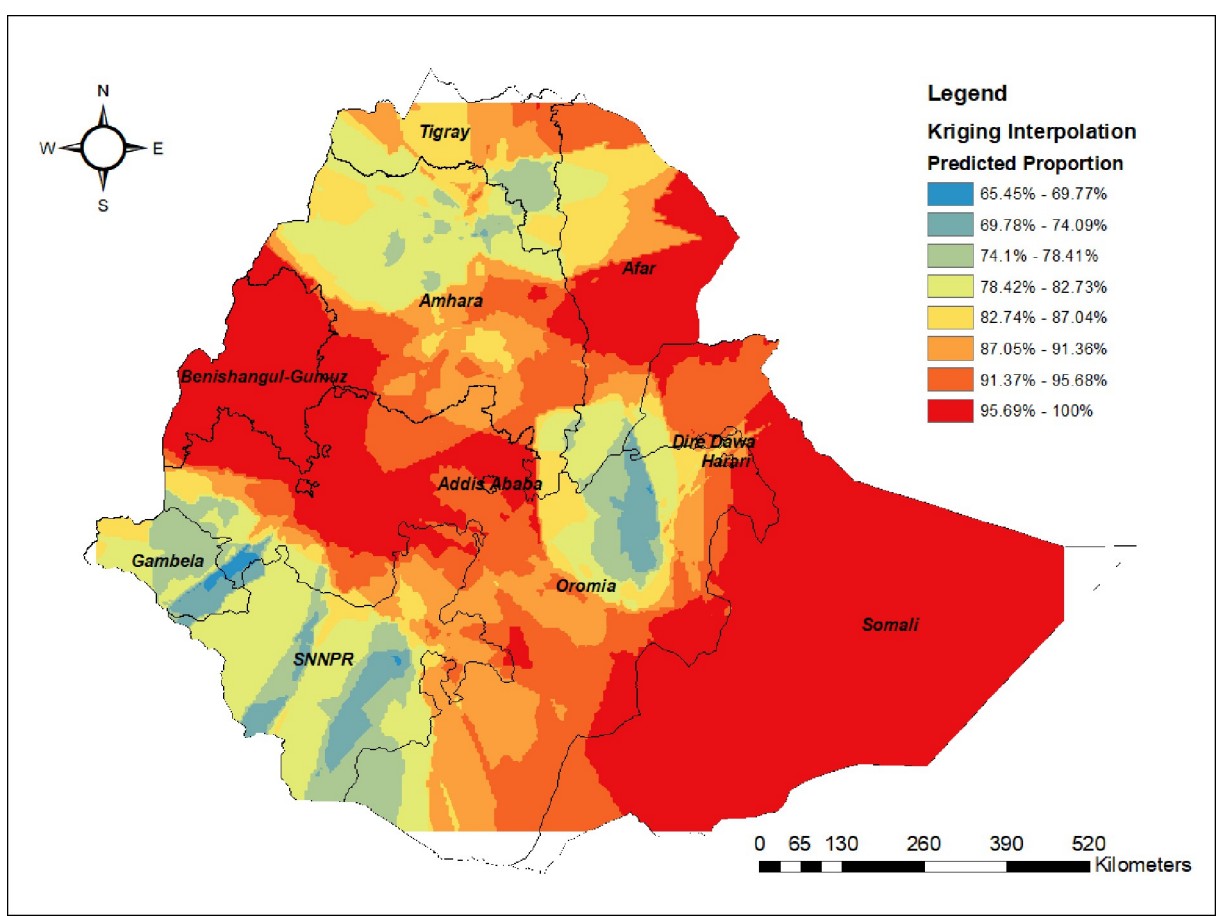

**Fig 3. Kriging interpolation of non-uptake of MCV2 in Ethiopia, Map produced using Arc GIS version 10.7.**

Children with a higher birth order were more likely of getting MCV2 as compared to the first birth order child. However, it is contrary to the finding of the studies conducted elsewhere on other types of vaccinations [30–32]. This may be because mothers with higher birth orders have practical knowledge of the benefits of vaccinations from preceding pregnancies or childbirths.

In addition, children who fully took the other basic vaccinations were more likely to receive MCV2 as compared to their counterparts. This is because mothers may have received health education and other services during their children's earlier vaccinations.

The spatial analysis revealed that Somalia, Benishangul-Gumuz, western Oromia, and central and southeastern Afar regions had a lower predicted proportion of MCV2 uptake. Based on the SaTScan analysis, the primary cluster was detected in the Benishangul-Gumuz region. The spatial variation among the regions might be due to the difference in access to health information, health facility, and socio-cultural difference in the settings. This regional variation of MCV2 uptake is also supported by other studies conducted in Ethiopia on the uptake of MCV1 [28].

The findings of the study could help policymakers and other responsible bodies to make appropriate interventions by focusing basic vaccine coverage and hotspot regions such as Benishangul-Gumuz region. Besides, it can be used as a baseline for future studies.

The study had many strengths. First, it is based on nationally representative data, and the data were weighted to restore representativeness and to get an appropriate statistical estimate.

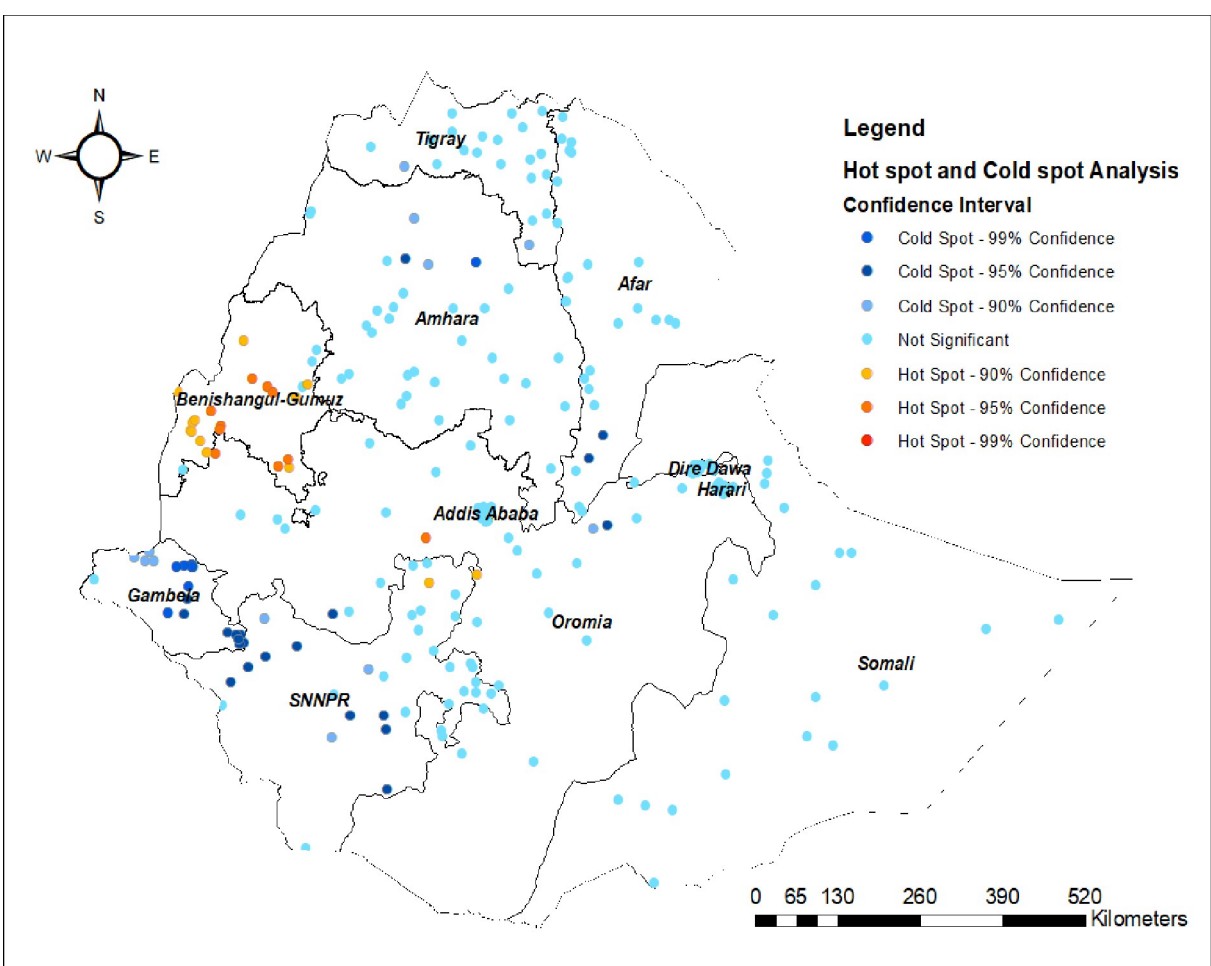

**Fig 4. Hot spot and cold spot analysis of non-uptake of MCV2 in Ethiopia, Map produced using Arc GIS version 10.7.**

Second, we have employed the multilevel analysis which is an appropriate model to consider the hierarchical nature of the data. Lastly, assessing the spatial variation is helpful for policy-makers to have a wise decision for appropriate intervention in light of limited resources. However, this study is not without limitation since it used the mini demographic and health survey data that misses some important variables such as distance from the health facility, women's decision-making autonomy, and facility-level factors. Besides, since the vaccine is recently introduced (the time between the introduction of the vaccine and data collection is short), the proportion of MCV2 uptake may be underestimated.

## Conclusion

The uptake of MCV2 in Ethiopia was low. A child from an older mother, a child with lower birth order and children who haven't fully taken other vaccines were less likely to take MCV2. Besides, MCV2 uptake was clustered in Ethiopia and the primary cluster spatial window was located in the Benishangul-Gumuz region. Therefore, the special concern should be given to regions with the lower uptake of MCV2 with prioritization for the Benishangul-Gumuz region. Besides, it is better to give special attention to basic vaccination programs.

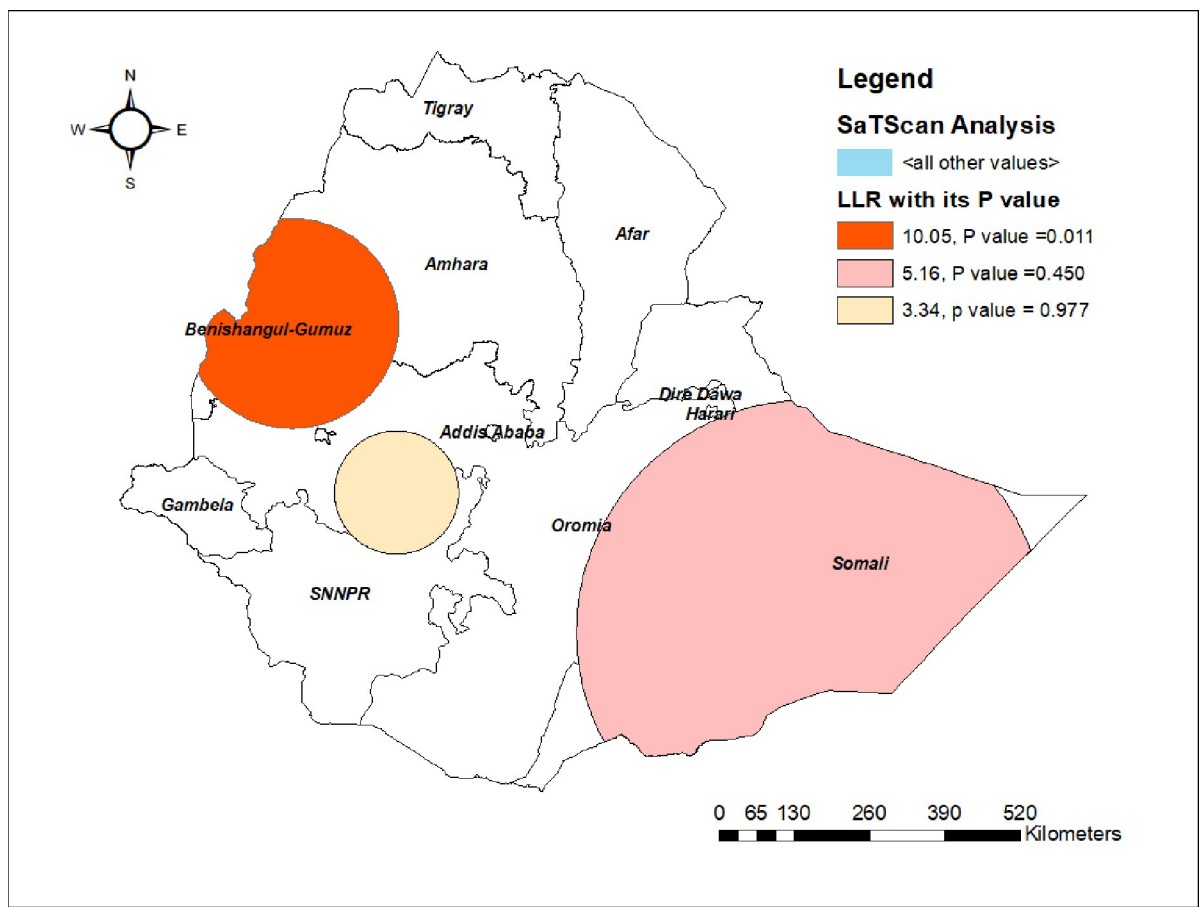

**Fig 5. SaTScan analysis of non-uptake of MCV2 in Ethiopia, Map produced using Arc GIS version 10.7 and SaTScan version 9.6.**

## Acknowledgments

Our heartfelt thanks go to the MEASURE DHS Program, which permitted us to use EMDHS data and GPS/coordinate files.

## Author Contributions

**Conceptualization:** Achamyeleh Birhanu Teshale.

**Data curation:** Achamyeleh Birhanu Teshale, Tsegaw Amare.

**Formal analysis:** Achamyeleh Birhanu Teshale, Tsegaw Amare.

**Investigation:** Achamyeleh Birhanu Teshale, Tsegaw Amare.

**Methodology:** Achamyeleh Birhanu Teshale, Tsegaw Amare.

**Resources:** Achamyeleh Birhanu Teshale, Tsegaw Amare.

**Software:** Achamyeleh Birhanu Teshale, Tsegaw Amare.

**Validation:** Achamyeleh Birhanu Teshale, Tsegaw Amare.

**Visualization:** Achamyeleh Birhanu Teshale, Tsegaw Amare.

**Writing – original draft:** Achamyeleh Birhanu Teshale, Tsegaw Amare.

**Writing – review & editing:** Achamyeleh Birhanu Teshale, Tsegaw Amare.

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
