## [Decision Letter · Decision Letter 0]

3 May 2022

PONE-D-22-01389Spatial distribution and factors associated with uptake of measles-containing second dose vaccine among children aged 24 to 35 months in EthiopiaPLOS ONE

Dear Dr. Teshale,

Thank you for submitting your manuscript to PLOS ONE. After careful consideration, we feel that it has merit but does not fully meet PLOS ONE’s publication criteria as it currently stands. Therefore, we invite you to submit a revised version of the manuscript that addresses the points raised during the review process.

We look forward to receiving your revised manuscript.

Kind regards,

Sanjay Kumar Singh Patel, Ph.D.

Academic Editor

PLOS ONE

Journal Requirements:

- https://bmcpublichealth.biomedcentral.com/articles/10.1186/s12889-019-7529-z

In your revision ensure you cite all your sources (including your own works), and quote or rephrase any duplicated text outside the methods section. Further consideration is dependent on these concerns being addressed.

3. We note that Figure 2, 3 and 4 in your submission contain map images which may be copyrighted. All PLOS content is published under the Creative Commons Attribution License (CC BY 4.0), which means that the manuscript, images, and Supporting Information files will be freely available online, and any third party is permitted to access, download, copy, distribute, and use these materials in any way, even commercially, with proper attribution. For these reasons, we cannot publish previously copyrighted maps or satellite images created using proprietary data, such as Google software (Google Maps, Street View, and Earth). For more information, see our copyright guidelines: http://journals.plos.org/plosone/s/licenses-and-copyright.

 a. You may seek permission from the original copyright holder of Figure 2, 3 and 4 to publish the content specifically under the CC BY 4.0 license. 

4. Please include a caption for figure 4.

Reviewers' comments:

Reviewer's Responses to Questions

**Comments to the Author**

1. Is the manuscript technically sound, and do the data support the conclusions?

Reviewer #1: Partly

Reviewer #2: Yes

2. Has the statistical analysis been performed appropriately and rigorously? 

Reviewer #1: Yes

Reviewer #2: Yes

3. Have the authors made all data underlying the findings in their manuscript fully available?

Reviewer #1: Yes

Reviewer #2: No

4. Is the manuscript presented in an intelligible fashion and written in standard English?

Reviewer #1: No

Reviewer #2: No

5. Review Comments to the Author

Reviewer #1: Review for PLOS ONE 28.01.2022

Paper tile: Spatial distribution and factors associated with uptake of measles-containing second dose vaccine among children aged 24 to 35 months in Ethiopia

FEEDBACK

General feedback

This manuscript has the potential to further contribute to understanding MCV2 vaccination inequalities in Ethiopia and ts findings could be used by policy makers to inform interventions aiming to improve MCV2 vaccination coverage in Ethiopia. The subject (MCV2 coverage) is of interest to many and can enrich the available literature on understanding MCV2 coverage determinants when published. The methods used are appropriate for the study. However, the manuscript requires a major revision including language editing before it will be suitable for publication. The data source for the manuscript seems inappropriate because data collection was started just one month after the introduction of MCV2 thereby likely biasing the true MCV2 coverage downwards.

Specific feedback

I think, at this stage, grouping my feedback into different themes would be more efficient than giving it line by line.

Language

This manuscript requires language editing to be suitable for publication.

Background

1. Mention the agenda referred in line 102.

2. I suggest reframing the objective (lines 103 – 104) to something like “…. to explore individual and contextual factors associated with the uptake of the second dose of a measles-containing vaccine among children aged 24 – 35 months in Ethiopia”.

Methods

1. Lines 109 – 110: Considering that routine MCV2 was introduced in Ethiopia in February 2019 and the data collection for the EMDHS (data source for this study) was conducted arch to June of the same year, it is possible that not all the children in the study sample had access to the vaccine before the data collection period. I suspect that this data source is inappropriate for objectives of this study as the time between the introduction of the vaccine and data collection are very close. Please indicate if the MCV2 dose was introduced through a campaign approach or otherwise. If it was not introduced trough a campaign approach, this limitation of the data source should be stated in the discussion/methodological considerations. If the vaccine was introduced in different regions in different dates, this should lo be indicated as it can also influence the coverage.

2. Line 129: I think the sample size (unweighted = 800 and weighted = 809) is referring to children 24 – 35 months not their mothers since it is the children’s dataset which is used.

3. Lines 136 – 142 (independent variables): I recommend mentioning how each variable was operationalized/categorized instead of the three indicated.

4. Lines 144 – 149 – Operational definition for region giving here is not the same as the one used in the spatial analysis. Going by the re-categorization explained here, one would expect a similar merging on the maps presented.

5. Lines 151 -155: Usually, the definition for full vaccination for basic vaccines is 1 dose of BCG, three doses of DPTCV/Penta, 3 doses of OPV, and 1 dose of MCV. I suggest using this definition since it is the same definition used in the EMDHS as can read from the report using the link below https://dhsprogram.com/pubs/pdf/FR363/FR363.pdf.

6. Line 161 and elsewhere: I suggest replacing “proportion of uptake of MCV2” with something like “MCV2 uptake” or “MCV2 coverage” or “proportion of children vaccinated with MCV2”, etc.

7. Lines 170 -171: I think this approach to variable selection is perhaps more useful for building predictive models. Though it is widely used, I think is better to include variables based on a priori and their availability in the dataset in this case because the significance level or effect of an independent variable in bivariable analysis may not be exactly the same when introduced in a multivariable regression model.

8. Lines 203 – 204: Perhaps it will be better to just mentioned that you used a secondary dataset and whether the EMDHS was ethically cleared.

Results

Line 221: Since vaccination card retention was reported to be low among the respondents for this study (26% - https://dhsprogram.com/pubs/pdf/FR363/FR363.pdf), I recommend doing a sensitivity analysis to assess the potential recall bias and report the results here and in the discussion.

Discussion

1. I suggest restructuring the discussion in the following order:

a. Summarize the key findings of the study in relation to the study objective.

b. Interpret (explain) the results taking into account the study objective, how the results compare with similar studies, etc.

c. Highlight the policy and research implications of the study.

d. Discuss the strengths and weaknesses of the study.

2. You may wish to consider including the following methodological limitations in your discussion:

a. The short time span between MCV2 introduction and the beginning of data collection for the EMDHS.

b. The potential of recall bias (depending on the results of the sensitivity analysis) considering the low card retention rate in the age cohort used in this study.

3. The explanation given in lines 289 – 292 for older mothers seems to be in conflict with the one given in lines 293 – 295 birth order number assuming that older mothers will generally have more children than younger mothers.

References

You may wish to consider the following points (they can be easily fixed):

1. Some of the references seem to be inappropriate where they are cited. For example,

a. References 8 & 9 would be better placed where reference 6 & 7 are because they are studies about inequalities in coverage but not deaths.

b. Reference 10 is about the prevalence of anemia but it is cited for how measles is spread.

2. Some references seem a bit too old to use. For example:

a. The refences (25 & 26) for MCV2 coverage in Africa are outdated in the sense that coverage has improved a lot now compared to the years referred (2014). That was at a time that many African countries had not introduced the second dose of the MCV2 dose.

Reviewer #2: Comments to Authors

Manuscript Number: PONE-D-22-01389

The authors investigated one of the packages included under primary health care services entitled “Spatial distribution and factors associated with uptake of measles-containing second dose vaccine among children aged 24 to 35 months in Ethiopia”. The issue has paramount importance to evaluate the national health sector commitments and SDGs. The authors also used a more of national representative data that may reflect inclusive picture of the nation.

General comments

1. The authors included children aged 24-35 months old. But the second dose of measles is given most of the time at the age of 15 months. Why they excluded children between 15-23 months of age as this could underestimate the vaccine uptake? They don’t have the confidence to say the intake is low and should reconsider this issue.

2. Background: This section is very lengthy, run out from the objective of the study. It lacks coherence and articulation of the research question. The author’s perspective may be better if they tried to see the existence of heterogeneity b/n individual, households and district or community level factors. There are also variables unforeseen and may have potential contributions for low intake of the MCV2.

3. Method: This section also needs substantial revision and clarity for readers. In some part the authors stated as the study is primary one but actually it is from dataset assembled for other purpose. Why the authors exclude one from two or more children per mother (may be twins or with very narrow birth interval). At all the conclusion is about children, excluding them from the analysis is irrational. The measurements of composite variables are not clearly mentioned. The criteria used for doing the random effect create confusion and not persuade for readers.

4. Results and discussion: The result section should be reorganized again by avoiding unnecessary narration and tabular representations. It should be without interpretations and methodological descriptions. The authors should move explanations of statistical analysis to the method section. The discussion is also not solid and analytical. The theoretical and practical implications of the main findings were not justified.

5. Overall the study needs extensive English language copy-editing.

---

## [Author Response · Author response to Decision Letter 0]

1 Oct 2022

Date: October 01, 2022

Authors response to the comments

Dear the Editor and the Reviewers, thanks for your comments and suggestions for the betterment of our manuscript. We have considered all your comments and suggestions in the revised manuscript as well as in the point by point response.

Response to comments on journal requirements 

Authors response: Amended according to the journal guideline

- https://bmcpublichealth.biomedcentral.com/articles/10.1186/s12889-019-7529-z

In your revision ensure you cite all your sources (including your own works), and quote or rephrase any duplicated text outside the methods section. Further consideration is dependent on these concerns being addressed.

Authors response: Thanks, we have considered it in the revised manuscript.

3. We note that Figure 2, 3 and 4 in your submission contain map images which may be copyrighted. All PLOS content is published under the Creative Commons Attribution License (CC BY 4.0), which means that the manuscript, images, and Supporting Information files will be freely available online, and any third party is permitted to access, download, copy, distribute, and use these materials in any way, even commercially, with proper attribution. For these reasons, we cannot publish previously copyrighted maps or satellite images created using proprietary data, such as Google software (Google Maps, Street View, and Earth). For more information, see our copyright guidelines: http://journals.plos.org/plosone/s/licenses-and-copyright.

a. You may seek permission from the original copyright holder of Figure 2, 3 and 4 to publish the content specifically under the CC BY 4.0 license.

Authors response: Thank you for the comment. The figures/maps are prepared using ArcGIS and SaTScan software after getting shape file from. https://spatialdata.dhsprogram.com/boundaries/#view=table&countryId=ET and coordinate files from the measure DHS program.

4. Please include a caption for figure 4.

Authors response: Thanks, we have added a caption for Figure 4.

Response to Reviewer #1 comments

The manuscript requires a major revision including language editing before it will be suitable for publication. The data source for the manuscript seems inappropriate because data collection was started just one month after the introduction of MCV2 thereby likely biasing the true MCV2 coverage downwards.

Authors response: Thank you very much for the important issue you raised. We have edited the over all manuscript. Besides, the data collection time was one month after the introduction of the vaccine and this is the limitation of the study and we put this as a limitation.

Background

1. Mention the agenda referred in line 102.

2. I suggest reframing the objective (lines 103 – 104) to something like “…. to explore individual and contextual factors associated with the uptake of the second dose of a measles-containing vaccine among children aged 24 – 35 months in Ethiopia”.

Authors response: Thanks. When we said the agenda, it was to mean MVC2 and now the overall background is modified accordingly. We also reframe the objective to read “to explore the spatial variations of the uptake of second dose measles-containing vaccine and to assess its individual and contextual factors among children aged from 24 to 35 months in Ethiopia” 

Methods

1. Lines 109 – 110: Considering that routine MCV2 was introduced in Ethiopia in February 2019 and the data collection for the EMDHS (data source for this study) was conducted march to June of the same year, it is possible that not all the children in the study sample had access to the vaccine before the data collection period. I suspect that this data source is inappropriate for objectives of this study as the time between the introduction of the vaccine and data collection are very close. Please indicate if the MCV2 dose was introduced through a campaign approach or otherwise. If it was not introduced trough a campaign approach, this limitation of the data source should be stated in the discussion/methodological considerations. If the vaccine was introduced in different regions in different dates, this should be indicated as it can also influence the coverage.

Authors response: Thank you. It is given as a campaign approach and involved in a standard care as well in every region. Besides, since the time between the introduction of the vaccine and data collection short/very close, we acknowledge this as our study’s limitation in the last paragraph of the discussion section. 

2. Line 129: I think the sample size (unweighted = 800 and weighted = 809) is referring to children 24 – 35 months not their mothers since it is the children’s dataset which is used.

Authors response: Thank you. We have amended it to read children. 

3. Lines 136 – 142 (independent variables): I recommend mentioning how each variable was operationalized/categorized instead of the three indicated.

Authors Response: Thank you. Majority of the variables are straightforward and did not require operational definition. However, we have putted the categories for each variable in brackets.

4. Lines 144 – 149 – Operational definition for region giving here is not the same as the one used in the spatial analysis. Going by the re-categorization explained here, one would expect a similar merging on the maps presented.

Authors response: Thank you for raising an important issue. In the multilevel analysis we categorized region in to three based on previous researches and for ease of intervention. We found that fitting without categorizing is not good since some regions have higher observation and others have very low samples per cell. However, for the spatial analysis which is descriptive study, we did not categorize it since our intention was to map the distribution or clustering of the uptake in the regions. Besides, it is impossible to do a spatial analysis using the re categorized region. 

5. Lines 151 -155: Usually, the definition for full vaccination for basic vaccines is 1 dose of BCG, three doses of DPTCV/Penta, 3 doses of OPV, and 1 dose of MCV. I suggest using this definition since it is the same definition used in the EMDHS as can read from the report using the link below https://dhsprogram.com/pubs/pdf/FR363/FR363.pdf. 

Authors response: Thank you very much for the comment. You are right the definition for full vaccination/immunization is as you stated. However, in our case, all children took the first dose of MCV, it is a prerequisite for the second dose. Therefore, we have removed it in the definition. 

6. Line 161 and elsewhere: I suggest replacing “proportion of uptake of MCV2” with something like “MCV2 uptake” or “MCV2 coverage” or “proportion of children vaccinated with MCV2”, etc.

Authors response: Thank you. We have amended it to read “MCV2 uptake”.

7. Lines 170 -171: I think this approach to variable selection is perhaps more useful for building predictive models. Though it is widely used, I think is better to include variables based on a priori and their availability in the dataset in this case because the significance level or effect of an independent variable in bivariable analysis may not be exactly the same when introduced in a multivariable regression model.

Authors response: Thank you. You are right. Considering your comment and the number of variables (limited number of variables) we conducted a re analysis and include all variables into the multilevel variables without considering the bivariable analysis.

8. Lines 203 – 204: Perhaps it will be better to just mentioned that you used a secondary dataset and whether the EMDHS was ethically cleared.

Authors response: Thank you. As you know DHS data is a well-known dataset, any information including ethical clearance is found in the Report (we put references in the method section under data source). But, our study used this easily accessible data set with no personal identifier inside. 

Results

Line 221: Since vaccination card retention was reported to be low among the respondents for this study (26% - https://dhsprogram.com/pubs/pdf/FR363/FR363.pdf), I recommend doing a sensitivity analysis to assess the potential recall bias and report the results here and in the discussion.

Authors response: Thank you. The recall bias is less likely since the time period between the uptake of the vaccine and the data collection is short. Besides, in this study, only 10 children (around one percent) had vaccination card and it is difficult to fit the model to conduct a sensitivity analysis (See Figure 1). 

Discussion

1. I suggest restructuring the discussion in the following order:

a. Summarize the key findings of the study in relation to the study objective.

b. Interpret (explain) the results taking into account the study objective, how the results compare with similar studies, etc.

c. Highlight the policy and research implications of the study.

d. Discuss the strengths and weaknesses of the study.

2. You may wish to consider including the following methodological limitations in your discussion:

a. The short time span between MCV2 introduction and the beginning of data collection for the EMDHS.

b. The potential of recall bias (depending on the results of the sensitivity analysis) considering the low card retention rate in the age cohort used in this study.

3. The explanation given in lines 289 – 292 for older mothers seems to be in conflict with the one given in lines 293 – 295 birth order number assuming that older mothers will generally have more children than younger mothers.

Authors response: Thank you. We have considered all the comments you raised in the discussion section.

References

You may wish to consider the following points (they can be easily fixed):

1. Some of the references seem to be inappropriate where they are cited. For example, 

a. References 8 & 9 would be better placed where reference 6 & 7 are because they are studies about inequalities in coverage but not deaths.

b. Reference 10 is about the prevalence of anemia but it is cited for how measles is spread.

2. Some references seem a bit too old to use. For example:

a. The refences (25 & 26) for MCV2 coverage in Africa are outdated in the sense that coverage has improved a lot now compared to the years referred (2014). That was at a time that many African countries had not introduced the second dose of the MCV2 dose.

Authors response: We have considered your comments on Reference in the revised manuscript accordingly.

Response to Reviewer #2 comments

The authors investigated one of the packages included under primary health care services entitled “Spatial distribution and factors associated with uptake of measles-containing second dose vaccine among children aged 24 to 35 months in Ethiopia”. The issue has paramount importance to evaluate the national health sector commitments and SDGs. The authors also used a more of national representative data that may reflect inclusive picture of the nation.

General comments

1. The authors included children aged 24-35 months old. But the second dose of measles is given most of the time at the age of 15 months. Why they excluded children between 15-23 months of age as this could underestimate the vaccine uptake? They don’t have the confidence to say the intake is low and should reconsider this issue.

Authors response: Thank you for your important comment. Yes, the second dose of measles vaccine is recommended to be administered for children starting from 15 months of age and we can include children started from age 15 months. But including a 15 months aged child to our study leads to biased estimation that may end up with underestimation of vaccine coverage if the child didn’t take the vaccine at 15 months. Because the child is still in the recommended age (15-23 year) to get vaccinated. 

2. Background: This section is very lengthy, run out from the objective of the study. It lacks coherence and articulation of the research question. The author’s perspective may be better if they tried to see the existence of heterogeneity b/n individual, households and district or community level factors. There are also variables unforeseen and may have potential contributions for low intake of the MCV2.

Authors response: we have considered your comments on the background section of the revised manuscript.

3. Method: This section also needs substantial revision and clarity for readers. In some part the authors stated as the study is primary one but actually it is from dataset assembled for other purpose. Why the authors exclude one from two or more children per mother (may be twins or with very narrow birth interval). At all the conclusion is about children, excluding them from the analysis is irrational. The measurements of composite variables are not clearly mentioned. The criteria used for doing the random effect create confusion and not persuade for readers.

Authors response: Thanks. We have amended the method section of the revised manuscript. Regarding selecting a child from a household with more than one child, the reason why we took the last child was since information such as ANC and other maternal characteristics are for the last child only.

4. Results and discussion: The result section should be reorganized again by avoiding unnecessary narration and tabular representations. It should be without interpretations and methodological descriptions. The authors should move explanations of statistical analysis to the method section. The discussion is also not solid and analytical. The theoretical and practical implications of the main findings were not justified.

Authors response: Thank you. We have revised manuscript according to the comments.

5. Overall, the study needs extensive English language copy-editing.

Authors response: we have edited the English write in the revised manuscript.

---

## [Decision Letter · Decision Letter 1]

20 Dec 2022

Exploring spatial variations and the individual and contextual factors of uptake of measles-containing second dose vaccine among children aged 24 to 35 months in Ethiopia

PONE-D-22-01389R1

Dear Dr.Teshale,

We’re pleased to inform you that your manuscript has been judged scientifically suitable for publication and will be formally accepted for publication once it meets all outstanding technical requirements.

Kind regards,

Tefera Chane Mekonnen, Master in Public Health(MPH)

Academic Editor

PLOS ONE

Additional Editor Comments (optional):

Reviewers' comments:

Reviewer's Responses to Questions

**Comments to the Author**

1. If the authors have adequately addressed your comments raised in a previous round of review and you feel that this manuscript is now acceptable for publication, you may indicate that here to bypass the “Comments to the Author” section, enter your conflict of interest statement in the “Confidential to Editor” section, and submit your "Accept" recommendation.

Reviewer #1: All comments have been addressed

Reviewer #2: All comments have been addressed

2. Is the manuscript technically sound, and do the data support the conclusions?

Reviewer #1: Yes

Reviewer #2: Yes

3. Has the statistical analysis been performed appropriately and rigorously? 

Reviewer #1: Yes

Reviewer #2: Yes

4. Have the authors made all data underlying the findings in their manuscript fully available?

Reviewer #1: Yes

Reviewer #2: Yes

5. Is the manuscript presented in an intelligible fashion and written in standard English?

Reviewer #1: No

Reviewer #2: No

6. Review Comments to the Author

Reviewer #1: Although my comments have been addressed, this manuscript would required proof reading before publication.

Reviewer #2: The authors have addressed my previous comments and the manuscript has been improved. It needs copyediting to make it more sound for readers of the scientific community.

7. PLOS authors have the option to publish the peer review history of their article (what does this mean?). If published, this will include your full peer review and any attached files.

Reviewer #1: No

Reviewer #2: **Yes: **Tefera Chane Mekonnen

---

## [Editor Report · Acceptance letter]

26 Dec 2022

PONE-D-22-01389R1 

Exploring spatial variations and the individual and contextual factors of uptake of measles-containing second dose vaccine among children aged 24 to 35 months in Ethiopia 

Dear Dr. Teshale:

I'm pleased to inform you that your manuscript has been deemed suitable for publication in PLOS ONE. Congratulations! Your manuscript is now with our production department. 

Kind regards, 

on behalf of

Dr. Tefera Chane Mekonnen 

Academic Editor

PLOS ONE